# The Effect of Heat Treatment of β-Tricalcium Phosphate-Containing Silica-Based Bioactive Aerogels on the Cellular Metabolism and Proliferation of MG63 Cells

**DOI:** 10.3390/biomedicines10030662

**Published:** 2022-03-12

**Authors:** Csaba Hegedűs, Zsuzsanna Czibulya, Ferenc Tóth, Balázs Dezső, Viktória Hegedűs, Róbert Boda, Dóra Horváth, Attila Csík, István Fábián, Enikő Tóth-Győri, Zsófi Sajtos, István Lázár

**Affiliations:** 1Department of Biomaterials and Prosthetic Dentistry, Faculty of Dentistry, University of Debrecen, 4032 Debrecen, Hungary; czibulya.zsuzsanna@dental.unideb.hu (Z.C.); ferenc.toth@med.unideb.hu (F.T.); 2Department of Oral Pathology and Microbiology, Faculty of Dentistry, University of Debrecen, 4032 Debrecen, Hungary; dezsob51@gmail.com; 3Department of Pediatric Dentistry and Orthodontics, Faculty of Dentistry, University of Debrecen, 4032 Debrecen, Hungary; hegedus.v123@gmail.com; 4Department of Oral and Maxillofacial Surgery, Faculty of Dentistry, University of Debrecen, 4032 Debrecen, Hungary; boda.robert@dental.unideb.hu (R.B.); horvath.dora@dental.unideb.hu (D.H.); 5Laboratory of Materials Science, Institute for Nuclear Research, Eötvös Loránd Research Network, 4026 Debrecen, Hungary; csik.attila@atomki.hu; 6Department of Inorganic and Analytical Chemistry, Faculty of Science and Technology, University of Debrecen, 4032 Debrecen, Hungary; ifabian@science.unideb.hu (I.F.); gyori.eniko@science.unideb.hu (E.T.-G.); sajtos.zsofi@science.unideb.hu (Z.S.)

**Keywords:** silica aerogel, aerogel composite, mesoporous material, β-tricalcium phosphate, bone regeneration, calvaria bone critical size defect model, remodeling

## Abstract

β-Tricalcium phosphate was combined with silica aerogel in composites prepared using the sol–gel technique and supercritical drying. The materials were used in this study to check their biological activity and bone regeneration potential with MG63 cell experiments. The composites were sintered in 100 °C steps in the range of 500–1000 °C. Their mechanical properties, porosities, and solubility were determined as a function of sintering temperature. Dissolution studies revealed that the released Ca-/P molar ratios appeared to be in the optimal range to support bone tissue induction. Cell viability, ALP activity, and type I collagen gene expression results all suggested that the sintering of the compound at approximately 700–800 °C as a scaffold could be more powerful in vivo to facilitate bone formation within a bone defect, compared to that documented previously by our research team. We did not observe any detrimental effect on cell viability. Both the alkaline phosphatase enzyme activity and the type I collagen gene expression were significantly higher compared with the control and the other aerogels heat-treated at different temperatures. The mesoporous silica-based aerogel composites containing β-tricalcium phosphate particles treated at temperatures lower than 1000 °C produced a positive effect on the osteoblastic activity of MG63 cells. An in vivo 6 month-long follow-up study of the mechanically strongest 1000 °C sample in rat calvaria experiments provided proof of a complete remodeling of the bone.

## 1. Introduction

After a long period of nonmedical use [1,2], aerogels have attracted interest in the biomedical field in the last decade [3,4,5]. Aerogels are lightweight materials with extremely high porosity and a large specific surface area [6,7,8]. Their material-specific biocompatibility [9,10] and biodegradability [11,12] make them promising materials for several applications, such as drug delivery [13,14,15,16] and tissue engineering [17,18].

Silica aerogel-based composites and hybrids were synthesized most recently in our laboratory. They are biocompatible materials that are appropriate for drug delivery [19,20] and bone replacement purposes [17,18,21,22]. Silicon, their basic element, in the form of water-soluble orthosilicate ions, was found to be an important factor in bone metabolism [23]. The orthosilicate stimulates the stem-cell differentiation [24], proliferation [25,26,27], and metabolic activity of osteoblast-like cells or osteoblasts [28,29], via the favorable modulation of osteoblast adhesion [30] and mineralization activity [31,32,33,34]. These changes involve the increased expression of insulin-like growth factor II (IGF-II) [31], osteocalcin, BMP-2, RUNX2, and type I collagen [35], associated with increased alkaline phosphatase (ALP) activity [36] and mineral deposition [37]. The effect of the orthosilicate depends on its concentration; at supraphysiological levels, it does not induce significant changes in the activity of bone-related cells [35], whereas an excessive amount might result in apoptosis [32] and decreased bone strength [38,39]. The biological function of silicon in Si-containing biomaterials, Si-based bioceramics [40], Si-substituted calcium phosphates [41], bioglasses [42,43], and silicon nitride [44] has been investigated. Among them, silicon nitride proved to be resistant to bacterial infection during bone regeneration, while the silicon content increased the osteogenic differentiation and proliferation of osteoblast-like cells [45]. A mixture of silicon- and calcium-containing biomaterials [46], as well as their combination with various other materials, including zinc and magnesium, improved antibacterial properties, gene expression, biological performance, and the differentiation and proliferation of osteoblasts [47,48,49,50,51].

β-Tricalcium phosphate (β-TCP) is currently one of the most commonly used calcium phosphates in bone tissue engineering [52]. Due to its advantageously high solubility combined with good biodegradation properties, most studies have used β-TCP over other TCPs [53]. As a consequence of the physiological conditions present, β-TCP can be transformed either partially or completely to hydroxyapatite (HAp), which is the major constituent of bones among the inorganic materials [54]. The role of β-TCP in the bones is to form a connecting bond at the interfaces of the host tissue [55,56]. The biocompatibility of β-TCP is excellent. It is not only biodegradable but also resorbable and osteoconductive. In addition, its Ca/P ratio is similar to that of bones [57,58,59], and it shows a moderate degradation rate when compared to HAp or α-TCP [60]. In studies performed with mesenchymal stem cells (MSCs), β-TCP promoted their attachment, proliferation, and differentiation [61,62]. Due to its excellent properties, β-TCP is now a recognized and widely used substitute material for bone tissue repairs [63,64,65,66].

The matrix component of our β-TCP-containing composites is the amorphous and extremely porous silica aerogel [6,67,68], which serves as the source of water-soluble orthosilicate ions. According to previous publications, the rate and extent of dehydration upon heat treatment, followed by the rehydration of amorphous silica surfaces in aqueous media, strongly depends on the temperature at which the sample was heat-treated [69,70]. Condensation processes of vicinal silanol groups, especially at high temperatures, result in the formation of new Si–O–Si covalent bonds; furthermore, their rehydration and the regeneration of silanol groups, which represent the first step of dissolution, are time-consuming processes [71,72].

We hypothesized, on the basis of our earlier studies [17,21,22,65], that our aerogel composites treated at different temperatures may provide the optimum release rate of calcium, phosphate, and orthosilicate ions and effectively stimulate osteogenic activity and bone healing. Hence, this study was aimed at preparing, characterizing, and systematically testing β-tricalcium phosphate–mesoporous silica aerogel composites treated at different sintering temperatures (600, 700, 800, 900, and 1000 °C), as well as investigating the effect of heating temperature on the dissolution rates of the chemical components and their connection with the osteoblastic activity of MG63 cells.

## 2. Materials and Methods

### 2.1. Synthesis of β-TCP-Modified Aerogels

All samples were prepared using the process described previously in the form of small cylinders of 5 mm OD and 5 mm height [65]. Heat treatment of the samples was performed in a WiseTherm programmable temperature furnace. First, all samples were heated at 500 °C for 8 h to bake out the disposable porogen material microcrystalline cellulose, and then selected samples were sintered in the range of 600–1000 °C in 100 °C steps, each for 1 h.

### 2.2. Nitrogen Adsorption–Desorption Porosimetry

Porosimetry studies were performed with a Quantachrom 2200e porosimeter (Quantachrom Instruments, Boynton Beach, FL, USA) at 77 K temperature, using nitrogen gas as the sorbate. The samples were ground to approximately 100 μm grains and degassed at 300 °C for 3 h prior to measurement. Then, 30 adsorption and 28 desorption data points were recorded for each sample. The isotherms were analyzed by the computer program NovaWin 11.0 (Quantachrom Instruments, Boynton Beach, FL, USA) using Brunauer–Emmett–Teller (BET), Barrett–Joyner–Halenda (BJH), and near-equilibrium density functional (DFT) theories. Statistical thickness values were calculated using the de Boer model, and the V–t plots were extrapolated to zero volume to estimate micropore contributions.

### 2.3. Grinding and Microscopic Study of β-TCP-Modified Aerogels (Aerogel Composites)

The composites heat-treated at different temperatures (600–1000 °C) were ground with an Analysette 3 Pro Vibratory Sieve Shaker plate, equipped with a ZrO_2_ mortar containing a 50 mm-diameter grinding ball. To reach appropriate homogeneity of the powders, a 1.5 mm amplitude was used for grinding. Each sample was treated for 2 min.

Samples for optical microscopy studies were ground by a vibratory shaker equipped with a stainless-steel ball mill (25 mm-diameter spherical mortar, with two 9.50 mm-diameter grinding balls, 5 mm amplitude, 5 min). In order to differentiate the silica aerogel matrix from the guest particles of β-TCP, the following staining protocol was followed: to 10–12 mg of ground powder, 20 μL of aqueous methylene blue solution (20 mg/mL) and 1.00 mL of water were added. After vigorously shaking for 1 min, the suspension was centrifuged at 13,800× *g* for 5 min. The aliquot was discarded, and the sediment was washed three more times with 1.00 mL of water. The residue was then suspended in 250 μL of water. For the microscopy study, a 50 μL portion was used. A Lacerta stereoscopic microscope with objectives of 4, 10, and 40× power was used, and the images were recorded using a 8 MPx camera (Hangzhou ToupTek Photonics Co. Ltd., Hangzhou, Zheijang, China); the scale was calibrated using an external calibration slide, and the images were analyzed by the ToupView 3.7 computer program (Hangzhou ToupTek Photonics).

### 2.4. Determination of the Compressive Strength Values

The compressive strength of the cylindrical samples was determined using a custom-made miniature mechanical tester tailored for the measurement of fragile and very sensitive aerogel samples. Specimens with two parallel faces were picked manually from a large number of particles. The load was changed in 0.1–1 N steps. Compressive strengths were calculated from the stress values belonging to the first fracture. At least five specimens were analyzed for each temperature.

### 2.5. Structural Study of as-Prepared and Heat-Treated Samples

To obtain crystallographic information from the as-prepared and annealed samples, X-ray diffraction (XRD) measurements were performed by a Rigaku SmartLab diffractometer (Rigaku Corporation, Akishima-shi, Tokyo, Japan) using Cu-Kα irradiation with wavelength λ = 0.154 nm. Scans were performed in θ–2θ scanning geometry, and the X-ray tube was operated at 200 mA and 45 kV. Diffraction patterns were measured in the range of 20–90° 2θ to characterize the crystalline phases in the samples. A step size of 0.01° 2θ per 4 s was applied in the process. Prior to measurement, the cylindrical samples were ground to a fine powder in a mortar and then pressed into the sample holder of the diffractometer.

### 2.6. Dissolution Dynamics and Ion-Release Properties of the Heat-Treated Composites

Due to the significant differences in the cell growth kinetics observed in the cell culture studies, the ion release dynamics of the composites were studied in stirred systems as a function of time. Equal amounts of ground composite aerogel materials (in large excess) were mixed in a constant volume of high-purity double deionized MilliQ water (Millipore SAS, 20, Molsheim, France) and stirred at a constant rate. Whole samples of the mixtures were taken after 15 min, 1 h, 1 day, 3 days, and 5 days, and then centrifuged; the supernatants were analyzed directly for Ca, P, and Si components by inductively coupled plasma–optical emission spectrometry (5110 ICP–OES, Agilent Technologies, Santa Clara, CA, USA).

Standard solutions were prepared from the mono-element spectroscopic standard of 1000 mg·L^−1^ stock solution (ICP IV, Merck, Kenilworth, NJ, USA). A five-point calibration process was used, for which standard solutions were diluted with 0.1 M HNO_3_ prepared in ultrapure water.

### 2.7. Cell Culture Studies

MG63 Sigma Aldrich (No. 86051601, St. Louis, MO, USA) cells were cultured in Minimum Essential Medium Eagle (Sigma Aldrich), supplemented with 10% (*v*/*v*) fetal bovine serum (Gibco, Waltham, MA, USA), 1% Glutamax (Gibco), and 1% antibiotic/antimycotic (Gibco), and incubated in a humidified incubator at 37 °C in 5% CO_2_. After reaching confluence, 4 × 10^4^ cells were seeded into 24-well plates and incubated under the same conditions. After reaching 100% confluence, the culture medium was replaced with the same media supplemented with 0.5 mg/mL of the appropriately heat-treated aerogel composite samples. Cells treated with culture medium without aerogel samples served as a negative control for the experiment. The culture medium was refreshed three times a week.

### 2.8. Scanning Electron Microscopy (SEM) Analysis of Structural Properties and Cell Growth

Scanning electron microscopy studies were performed using a Hitachi S-4300 instrument (Hitachi Ltd., Tokyo, Japan) equipped with a Bruker energy-dispersive X-ray spectrometer (Bruker Corporation, Billerica, MA, USA). High-resolution images were taken at accelerating voltages of 10 kV and 15 kV. The cell samples were sputter-coated with gold for 30 s. The aerogel samples required prolonged sputtering and were individually checked for custom coating. The structure of the aerogel composite samples was analyzed using fresh fracture surfaces.

Cells were grown in 24-well plates with the ground aerogel samples for 3 days to examine the nature of the interaction between the cells and aerogel composite particles. After the incubation period, the samples were fixed with 2% (*v*/*v*) glutaraldehyde solution for 2 h, and then with 1% OsO_4_ solution for 1 h. After fixation, the samples were dehydrated using graded ethanol solutions (10, 30, 50, 70, 80, 90, and 100% (*v*/*v*), 15 min each) and stored in absolute ethanol until examination. The dehydrated samples were critical-point-dried using CO_2_.

### 2.9. Cell Viability Assay

The viability of the cells was determined using the Alamar Blue assay (Thermo Scientific, Waltham, MA, USA). A total of 4 × 10^4^ cells/well were transferred to a 24-well plate and incubated at 37 °C and 5% CO_2_. Three parallels were set for each group. After reaching 100% confluence, the cells were treated with the appropriate aerogel samples with a concentration of 0.5 mg/mL. After 14 days, the cell culture medium of each well was replaced with 10% (*v*/*v*) Alamar blue reagent. After 1 h of incubation at 37 °C in the dark, the fluorescence of the samples was measured by a microplate reader (HIDEX Sense, Turku, Finland) at 544 nm excitation/595 nm emission.

### 2.10. Alkaline Phosphatase Activity Assay

Cells were seeded into a 24-well plate at a density of 4 × 10^4^ cells/well. Three parallels were set for each group. After reaching 100% confluence, the cells were treated with the appropriate composite samples at a concentration of 0.5 mg/mL. On day 14, the medium was removed, and the cells were lysed by an ALP lysis buffer as described earlier [65]. First, the cellular debris was separated by centrifugation. Then, the supernatant was mixed with 100 μL of *p*-nitrophenyl phosphate solution (SIGMAFAST™ *p*-Nitrophenyl phosphate Tablets; Sigma Aldrich), incubated overnight at room temperature, and analyzed at 405 nm. The total protein concentration of the lysate supernatants was measured by the BCA Protein Assay Kit (Thermo Scientific) and used for the normalization of ALP activity.

### 2.11. Gene Expression Analysis

For these measurements, the cells were seeded in triplicate as described above. After 14 days, the total RNA content of the cells was isolated by the Quick-RNA MiniPrep kit (Zymo Research, Irvine, CA, USA) according to the manufacturer’s instructions. Then, cDNA was synthesized from 1 µg of total RNA using a High-Capacity cDNA Reverse Transcription Kit (Applied Biosystems, Bedford, MA, USA). Type I collagen gene expression of the samples was measured by an Applied Biosystems Taqman Assay (COL1A1, Hs00164004_m1) and 5× HOT FIREPol Probe qPCR Mix Plus (no ROX) (Solis BioDyne, Tartu, Estonia), normalized by the GAPDH (glyceraldehyde 3-phosphate dehydrogenase, Hs02758991_g1) expression.

### 2.12. Statistical Analysis

Each experiment was carried out in triplicate. All the results in this study are reported as the mean ± standard deviation (SD). Experimental data were analyzed by Student’s t-test using the GraphPad Quick Calcs free web calculator (http://graphpad.com/quickcalcs/ttest2, accessed on 8 February 2022).

## 3. Results

### 3.1. Structural Studies

The shrinking of the cylindrical monolithic specimens as a function of the sintering temperature is shown in Figure 1. The native 25 °C sample contained the disposable porogen material microcrystalline cellulose, which was burnt out at 500 °C in the next step. Due to the high-temperature resistance of the silica aerogel matrix, significant shrinking started only at around 900 °C and became dominant above 1000 °C (not shown).

Heat treatment of the native (25 °C) composite resulted in samples with shrinking sizes and increasing compressive strength values. The diameters of the samples treated at different temperatures were as follows (in mm): 3.85 ± 0.05, 3.81 ± 0.07, 3.74 ± 0.10, 3.61 ± 0.05, 3.32 ± 0.07, and 2.15 ± 0.06 for the 500 °C, 600 °C, 700 °C, 800 °C, 900 °C, and 1000 °C samples, respectively. The compressive strength values of the heat-treated samples were as follows (in MPa): 0.47 ± 0.11, 0.75 ± 0.10, 0.68 ± 0.14, 0.59 ± 0.05, 1.01 ± 0.16, and 16.3 ± 4.28 for the 500 °C, 600 °C, 700 °C, 800 °C, 900 °C, and 1000 °C samples, respectively.

Particle sizes play an important role in all cell studies. Therefore, we set out to study the efficiency and effect of ball grinding on the particle size and structure. The optical microscopy study revealed that the silica–TCP composites suffered matrix cracking, as well as debonding of the TCP particles on strong mechanical impact (Figure 2). It was observed that the degree of degradation strongly depended on the temperature of heat treatment, which determined the hardness of the samples. As the pristine silica aerogel and the 500 °C sample were light and relatively flexible materials, they suffered much less fragmentation upon ball grinding than the 800 and 1000 °C samples did. The latter sample had the highest hardness and was quite brittle upon strong mechanical impact.

Structural changes upon heating may have a strong impact on the strength, porosity, surface polarity, and solubility of the composite samples. Scanning electron microscopy studies provided detailed insight into the fine structure of the aerogel matrix regions (Figure 3). The SEM images of all samples showed typical sol–gel silica aerogel structures with visible differences mainly in the interparticle spaces. Figure 3 shows that the globules did not melt or coalesce even at the highest temperature, indicating that no low melting phase was formed from the silica and β-TCP in the sintering process. The effects of the sintering temperatures are better reflected in the porosimetry results (Table 1).

The surface properties, polarities, pore size distribution (Figure 4), and the internal void space (cumulative pore volume) of the heat-treated samples were determined by nitrogen adsorption–desorption porosimetry at 77 K, and the data are summarized in Table 1. The adsorption isotherms showed type IV curves characteristic of mesoporous materials.

Table 1 shows the data of pore size distribution and pore volumes. All the BET surface areas were quite high; even the 1000 °C sample had a high specific surface area. These high values are typical for silica aerogels, and the contribution of the guest particles was not significant. The BET surfaces reached the maximum at 600 °C, as a result of the complete burning out of the disposable porogen cellulose residues. At higher temperatures, the surface area decreased with the shrinking of samples.

The C-constant, which shows the polarity of the surface, was 82.46 for the 500 °C sample. This value is typical for native silica aerogels. Heating results in a gradual loss of water by the condensation of surface silanol groups, leading to a decrease in the C-constant. All the 600–1000 °C samples had a medium-polarity surface. Due to the very high porosity and low density of the aerogel matrices, all samples had a very high cumulative pore volume, and the micropore contribution was negligible. The average pore diameter showed a maximum at 900 °C. However, this average value did not accurately reflect the structural changes. The observed maximum in pore diameter was a result of two changes. The pore size distribution curves in Figure 4 show the bimodal pore size distribution in the 500 °C and 600 °C samples. The maximum at approximately 20 nm refers to the porosity of the compactly packed domains of the globules, while the 40–50 nm shoulder refers to the interdomain spaces. With the increase in temperature, the globules gradually shrank. The more compact structure of the domains resulted in a transient widening of the interdomain spaces. Together with the overall shrinking of the skeleton, this resulted in a gradual shift in the maximum of the curves to higher diameters. However, at 1000 °C, the shrinking of the entire structure became so extensive that we could no longer differentiate between the types of pores.

Cell studies indicated an optimum heat-treatment temperature of 700–800 °C for the maximum biological effect. In order to find a connection between the solubility of the components of the aerogel composites and the extent of the biological effects, as well as to find the optimal preconditioning time, a dissolution/leaching study was performed. The chemical composition and concentrations of the leached elements (in ionic form) were monitored for 5 days. Figure 5 shows the dynamic change in concentration of the soaking solutions as a function of time.

Figure 5 shows that the dissolution of tricalcium phosphate was significantly faster in the first hour than that of the silica aerogel matrix. In addition, the calcium and phosphate concentrations were much less dependent on the heat-treatment temperature than that of the silicate ions. The mobilization of silicate ions is a crucial factor regarding the biological activities. Samples from the 500–800 °C range exhibited nearly the same dissolution profile. However, the rate of silicate mobilization decreased significantly at and above 900 °C, whereby saturation did not occur even after 5 days. At such a high temperature, the dehydration of surface silanol groups was significant. The dissolution rate of the amorphous silica aerogel matrix was correlated with the degree of surface hydration. This requires the rehydration of the Si–O–Si surface groups and a change in the geometry, which are slow processes [73]. The surface dehydration and decrease in surface polarity were well documented by the two lowest C-constants shown in the porosimetry studies.

Calculation of the molar ratios of calcium, phosphate, and silicate ions offered additional insight into the effect of heat transformation on the aqueous dissolution mechanism. As shown in Figure 6, the Ca/P molar ratio did not reach the theoretical value of 1.5 characterizing the chemical formula of TCP (Ca_3_(PO_4_)_2_) even after 5 days, while it approached a near-saturation value. The virtual phosphate deficiency was compensated for by silicate ions and finally reached an equilibrium value of approximately 0.05 Ca/Si, indicating that a small extent of calcium ions were mobilized in the form of calcium silicate. The actual values of Ca/Si and P/Si were nearly identical except in the 1000 °C sample, where they were significantly higher. This change is in good agreement with the reluctant dissolution of the silicate ions at that temperature, as discussed earlier. On the other hand, however, a set of complex solution-phase equilibria could have also contributed to the observed ionic concentrations and molar ratios due to the fact that the pH was not buffered. Buffering was excluded in order to avoid the modification of the dissolution profile of silica as a consequence of the presence of sodium ions [74,75]. Obviously, differently protonated species of phosphate ions (mainly monohydrogen and dihydrogen phosphates) were present in the solution in equilibrium with each other [76].

Figure 7 shows the XRD results obtained for the as-prepared and annealed samples. The Powder Diffraction Files (PDF) database was used for phase identification. The phase was determined using standard ICDD (International Center for Diffraction Data) cards. As can be seen, the diffraction pattern of TCP powder exactly matched card number #00-055-0898, confirming the application of tricalcium phosphate. The result of the measurement of the aerogel sample showed a typical amorphous pattern characteristic of the silica aerogel material. From the group of heat-treated samples, three samples were selected for diffraction measurements. The results revealed that annealing did not result in significant changes in the structure of β-tricalcium phosphate, which maintained its crystalline structure during annealing up to 1000 °C. This result is in good agreement with the literature, where earlier studies reported that β-tricalcium phosphate can only transfer to the α- and γ-phases at temperatures in excess of 1150 °C and 1430 °C, respectively [66].

### 3.2. Cell Viability Assay

To study whether the aerogel composites had any effect on the viability or proliferation of the cells, they were grown in the presence or absence (control) of the differently heat-treated aerogels and examined after 14 days of culture using the Alamar Blue assay. The cells cultured with the 700 and 600 °C heat-treated aerogel composite samples showed a slight but significant reduction in their metabolic activity compared to the control (Figure 8). However, the metabolic activity of the cells grown with the other composite samples was also reduced compared to the control; these alterations proved to be not significant.

### 3.3. Alkaline Phosphatase Activity Assay

The alkaline phosphatase (ALP) activity of cells was measured after 14 days of incubation with the differently heat-treated aerogel composites. After this incubation period, a significant increase was observed in the ALP activity of the cells incubated in the presence of any of the composites compared to the control cells (Figure 9). Moreover, the comparison of the ALP activity of cells cultured with the aerogel composites heat-treated in different temperatures showed that the activity in the cells grown with the aerogel composites heat-treated at 900 and 800 °C was slightly but significantly higher than that in the case of the 1000 °C heat-treated sample. The other comparisons between samples showed no more significant differences as a function of heat-treatment temperature.

### 3.4. Gene Expression Analysis

The gene expression of type I collagen in the cells grown in the presence or absence (control) of the differently heat-treated aerogels was examined after 14 days of culture. Type I collagen gene expression was significantly upregulated in the cells grown with the aerogels that were heat-treated at 700 °C and 800 °C, while it was significantly downregulated where the cells were grown with the aerogels heat-treated at other temperatures (Figure 10), compared to the untreated control.

## 4. Discussion

Silicon (Si)-based products enhance the osteogenic differentiation and proliferation of osteoprogenitor cells, and Si can directly influence bone quality. One of the significant characteristics of silicate-based inorganic biomaterials is that they can induce the formation of bone-like apatite, which plays an important role in enhancing the bonding between the implants and human host bone. The combination of bioactive elements (Sr, Zn, and Si or Sr, Mg, and Si) in bioceramics is a viable approach to improve the biological performance of biomaterials.

Our earlier study already demonstrated the applicability of monolithic mesoporous composite aerogel material sintered at a temperature of 1000 °C as a scaffold in rat calvaria critical size defect experiments (University of Debrecen Institutional Animal Care and Use Committee, no. 7-M/15/DE MÁB) [65].

Figure 11 shows images of the representative hematoxylin–eosin (HE)-stained tissue sections of formalin-fixed and decalcified samples obtained from the critical size model on rat calvaria bone with 6 mm round-shaped artificial parietal defects that were filled with the β-TCP–aerogel composite (1000 °C) to facilitate the process of re-ossification for repair. Figure 11A shows that after 1 month of the bone defect being filled with the aerogel material, there was active inflammation with the presence of exogenous compounds (blue arrows, blue asterisk) and fibrous granulation tissue formation. Within the solidified fibrous inflammatory granulation tissue, foci of small calcifications and early ossifications were recognized, resulting in the formation of immature bone (IB) islets inside the bone defect surrounded by osteoblasts (black arrows). The thick arrow points to the multinuclear macrophage in association with the exogenous material β-TCP–aerogel (also called a foreign body giant cell). In Figure 11B, 3 months after interaction of the defect with the β-TCP–aerogel, new bone (NB) formations progressed exponentially at multiple sites, each surrounded by an increased number of osteoblasts (black arrows). This reflects the powerful osteoinduction of β-TCP–aerogel to facilitate bone remodeling. After 6 months of the bone defect in the presence of β-TCP–aerogel (Figure 11C), re-ossification appeared nearly completed with features of a mature bone matrix and a decreased number of osteoblasts (NB), although some fibrous tissue area could still be recognized with the presence of remnants of exogenous β-TCP–aerogel particles that were not metabolized by this time (not shown). White dashed lines (left-hand side of Figure 11B,C) indicate the interface border of the original bone (edge of the defect) and the newly developed bone that became integrated.

Mechanical testing revealed that, although the compressive strengths did not change linearly with the temperature, a very sharp increase occurred at 1000 °C. This offers the possibility of custom machining and milling, as well as the application of such samples, even in load-bearing positions. The behavior of the composites upon impact and the friction associated with ball grinding showed a bipolar nature. The TCP particles liberated from the matrix upon grinding suffered the same degree of degradation independently of the temperature. However, the final particle size of the remaining composites differed significantly. Specimens treated at lower temperature suffered less degradation, demonstrating that the grinding properties are primarily determined by the elasticity of the matrix.

Porosimetry results showed that the characteristic pore sizes (i.e., the diameters belonging to the maximum of the pore-size distribution curves) were in the range of 20–60 nm, and the 500 and 600 °C curves exhibited a shoulder indicating a bimodal distribution. The advantage of the open pore structure and the wide pores on the molecular scale is that they provide a dynamically accessible diffusion-controlled channel system, which allows the transport of nutrients, dissolved gases, and metabolites upon contact with living cells.

The dissolution study helped us to answer the question of whether there is a correlation between the ratio of soluble chemical components, especially the Ca/P ratio characteristic of human bones, and the cell viability and different enzyme activities. The bones of healthy humans, depending on the age, sex, and type of bone, show a 1.60–1.80 ± 0.25 Ca/P molar ratio as calculated from the mass ratios given in [57,58,59,77]. A previous study demonstrated that calcium phosphates prepared at different molar ratios and sintered at 1100 °C show advantageous activities in osteoblast viability, collagen synthesis, ALP activity, and NO production in the range of 1.0–2.0 molar ratios. These parameters also strongly dependent on, among others, the particle size, pore size, and porosity [77]. As Figure 6 shows, the Ca/P ratio increased over time and stabilized around 1.2–1.3. This is in the right range to provide the necessary level of calcium and phosphate ions to exhibit the required biological activity, in accordance with the literature values, which supports the results of our present study. The less-than-theoretical Ca/P molar ratio indicates that some of the calcium ions were released in the form of calcium orthosilicate, which could be a consequence of the formation of a thin amorphous phosphate glass layer on the contacting surfaces [78,79]. However, the extent of this layer may not be significant, and no new crystalline phase or transformation of the beta-phase of TCP could be found in the XRD studies.

Regarding the biological activity of the samples, the specific surface area and the pore diameters played an important role, in accordance with the literature study. The specific surface area of 383 ± 1 m^2^/g and the pore diameter of 33 ± 1 nm of our materials seemed to be optimal for the enhanced cellular activities.

The examination of the cells grown in the presence of aerogel composites showed moderate reductions in cell viabilities at the lowest (600 and 700 °C) heat-treatment temperatures, while the cells treated with other aerogels also showed some reduction in their viability compared to the control, but these reductions proved to be not significant.

The alkaline phosphatase activity of the cells incubated with the aerogel composites was significantly increased in all cases compared to that of the control. Moreover, the comparison between aerogel composite heat-treatment temperatures showed that the ALP activity of the cells grown with the 800 and 900 °C heat-treated samples was even significantly higher than that of those grown with the 1000 °C sample.

The study of the effect of the differently heat-treated aerogels on the gene expression of type I collagen showed a very significant upregulation in the case of the cells treated with the 700 and 800 °C heat-treated aerogels, while the other aerogel composites significantly downregulated the expression of this gene.

## 5. Conclusions

The two important bioactive inorganic components required for bone regeneration, namely, β-tricalcium phosphate and water-soluble orthosilicate, were combined in this study in a special synthetic way to provide β-TCP–silica aerogel composites to check for their biological activity with MG63 cells and in vivo rat calvaria critical size defect experiments. A significant advantage of β-TCP–silica aerogel composites is that their physical strength and porosity can be tuned by the sintering temperature, which allows us to tailor the properties for a specific application. Dissolution studies revealed that the released Ca/P molar ratios were in the optimal range to support bone remodeling and that the dynamics of silica dissolution strongly depended on the sintering temperature.

From the evaluation of cell viability, ALP activity, and gene expression, it was concluded that the appropriate heat-treatment temperature of β-TCP–silica aerogel composites for the acceleration of bone formation showed an unexpected sharp maximum at around 800 °C, where we could not observe any detrimental effect on cell viability, but where both alkaline phosphatase activity and type I collagen gene expression were significantly higher compared with the control and the other aerogels heat-treated at different temperatures. The mesoporous silica-based aerogel composites containing β-tricalcium phosphate particles treated at or less than 1000 °C produced a positive effect on the osteoblastic activity of MG63 cells. An in vivo 6 month-long follow-up study of the mechanically strongest 1000 °C sample in rat calvaria critical size defect experiments provided proof of a complete remodeling of the bone.

Our experiments clearly demonstrated that the β-TCP–mesoporous silica aerogel composites stimulated osteogenic activity and bone healing; hence, they could find application as new experimental bone-substitute bioactive materials in dentistry and orthopedics in the future.

## Figures and Tables

**Figure 1 biomedicines-10-00662-f001:**
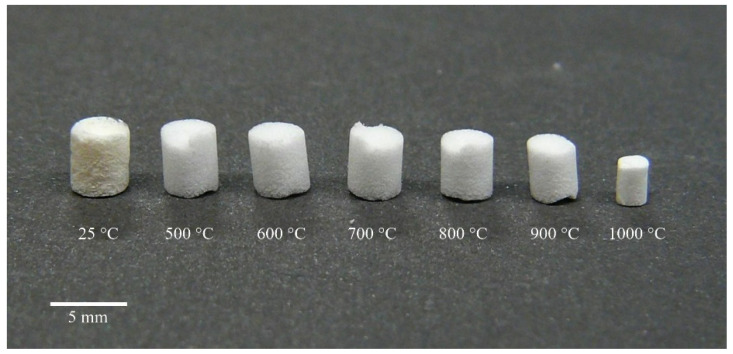
Demonstration of the effect of sintering temperature on the dimensions of cylindrical specimens. Scale bar: 5 mm.

**Figure 2 biomedicines-10-00662-f002:**
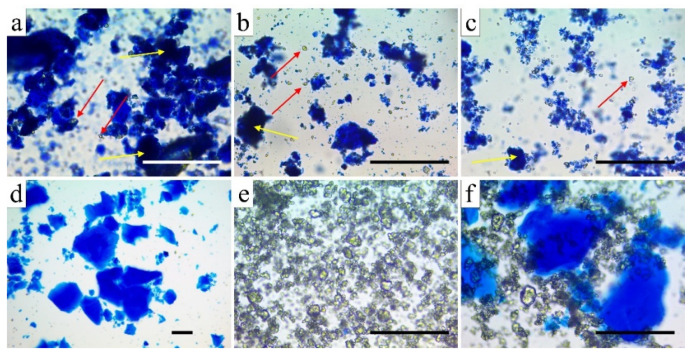
Optical microscopy images of ground aerogel composites, pristine silica aerogel, and β-TCP after methylene blue selective staining of the silica matrix. Red arrows indicate the largest ground and unstained β-TCP particles; yellow arrows show the largest intact composite regions. Scale bars: 100 μm. (**a**) Aerogel composite 500 °C; (**b**) aerogel composite 800 °C; (**c**) aerogel composite 1000 °C; (**d**) pristine silica aerogel; (**e**) commercial β-TCP; (**f**) mechanical mixture of pristine silica aerogel and β-TCP.

**Figure 3 biomedicines-10-00662-f003:**
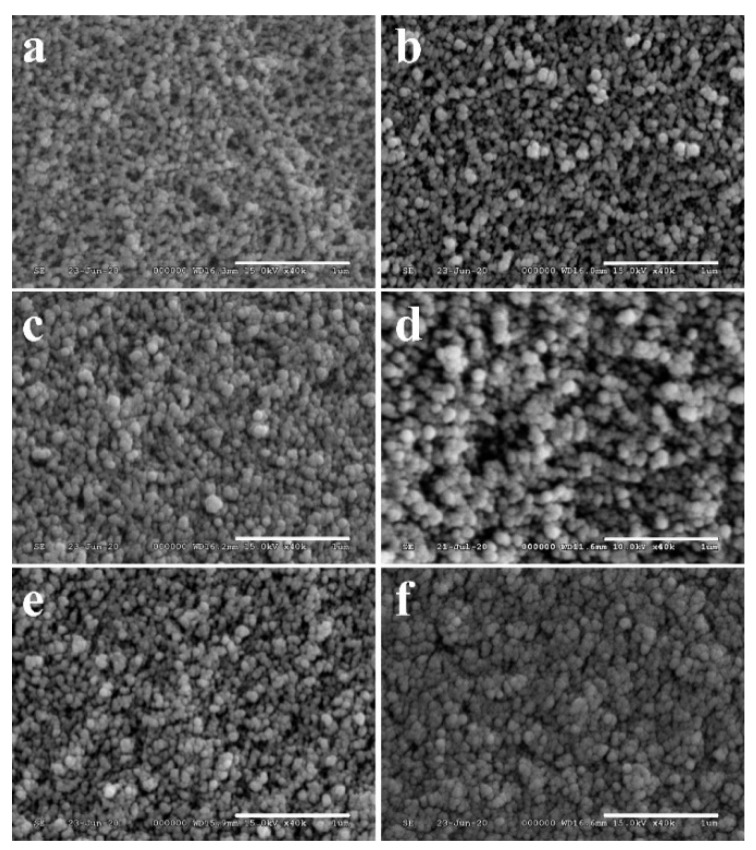
Scanning electron microscopy images of the structure of aerogel matrices in the composites calcined at different temperatures: (**a**) 500 °C; (**b**) 600 °C; (**c**) 700 °C; (**d**) 800 °C; (**e**) 900 °C; (**f**) 1000 °C. Scale bars: 1 μm.

**Figure 4 biomedicines-10-00662-f004:**
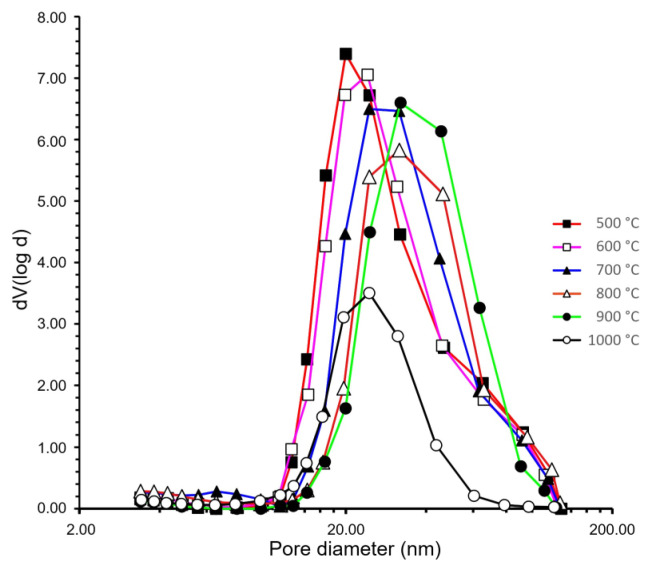
Pore size distribution curves of heat-treated samples calculated using the BJH method.

**Figure 5 biomedicines-10-00662-f005:**
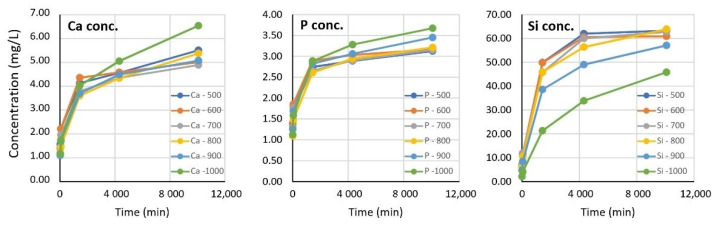
Dynamic change in concentration of the leached components as a function of time and heating temperature.

**Figure 6 biomedicines-10-00662-f006:**
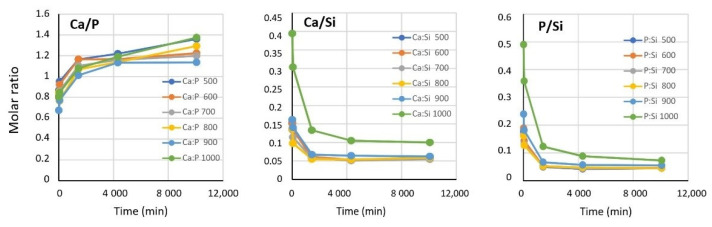
Molar ratios of the dissolved ionic species as a function of leaching time.

**Figure 7 biomedicines-10-00662-f007:**
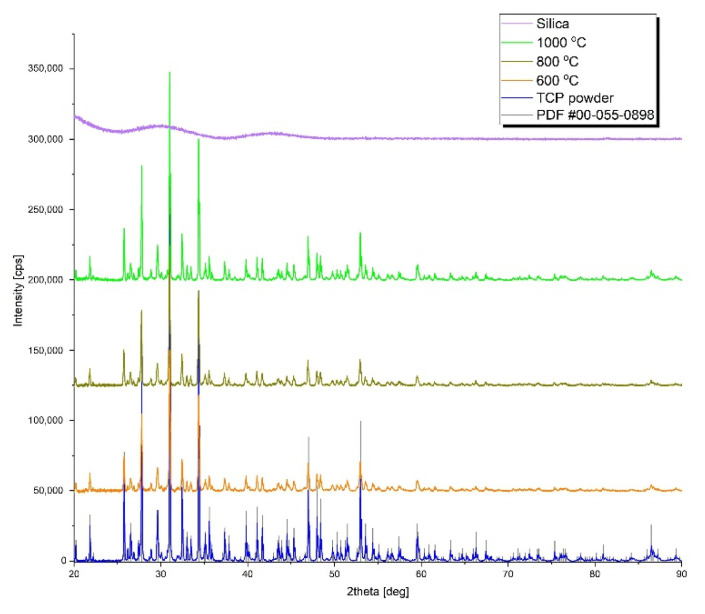
X-ray diffraction patterns of silica aerogel and β-TCP-modified aerogels before and after heat treatments. The positions of diffraction peaks from the PDF database (#00-055-0898) are also shown.

**Figure 8 biomedicines-10-00662-f008:**
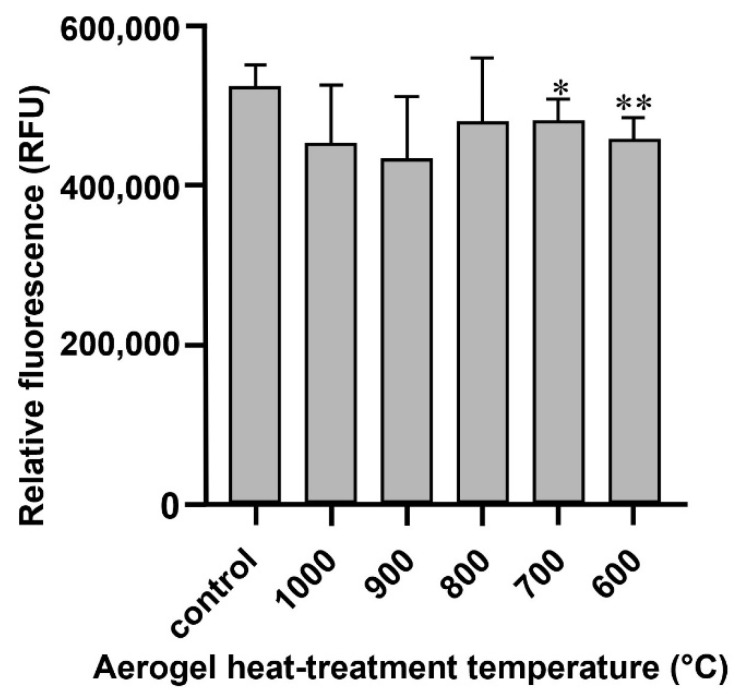
The effects of the differently heat-treated aerogel composites on the viability of MG63 cells cultured for 14 days as measured by the Alamar Blue assay. Values are expressed as sample means; error bars represent the estimates of standard deviations calculated from three parallel measurements compared to the control (* *p* < 0.05, ** *p* < 0.01).

**Figure 9 biomedicines-10-00662-f009:**
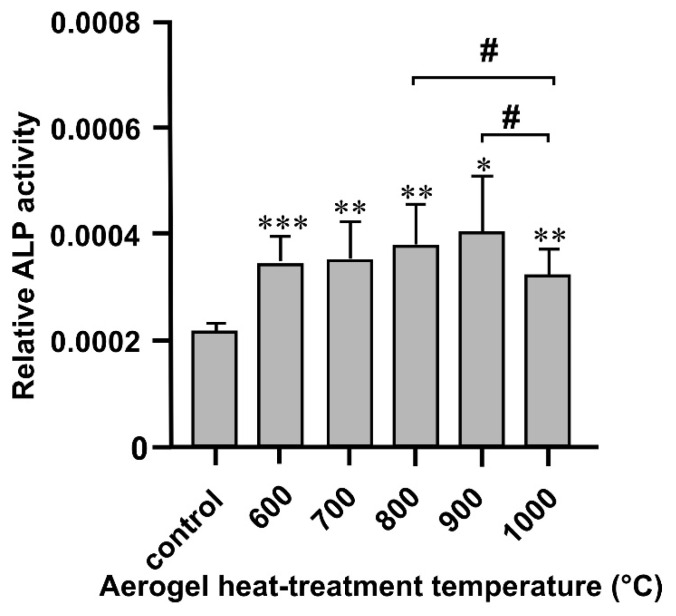
The effects of the differently heat-treated aerogel composites on the ALP activity of MG63 cells cultured for 14 days. Values are expressed as sample means; error bars represent the estimates of standard deviations calculated from three parallel measurements compared to the control (* *p* < 0.05, ** *p* < 0.01, *** *p* < 0.001); ^#^
*p* < 0.05 between the different temperatures.

**Figure 10 biomedicines-10-00662-f010:**
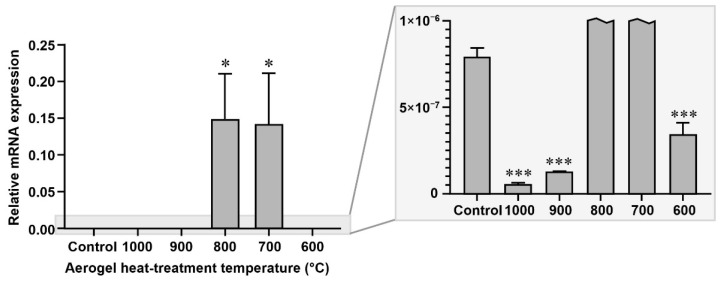
The effects of the differently heat-treated aerogel composites on the type I collagen mRNA expression of MG63 cells cultured for 14 days as measured by RT-qPCR. Values are expressed as sample means; error bars represent the estimates of standard deviations calculated from three parallel measurements compared to the control (* *p* < 0.05, *** *p* < 0.001).

**Figure 11 biomedicines-10-00662-f011:**
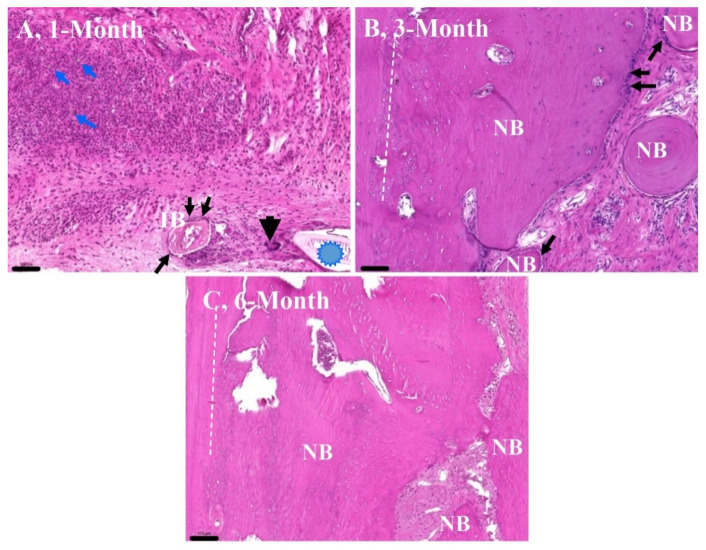
Images of the representative hematoxylin–eosin (HE)-stained tissue sections of formalin-fixed and decalcified samples obtained from the rat critical size defect model on rat calvaria bone. Magnifications and scale bars for all images are 100× and 100 μm, respectively. IB, immature bone; NB, new bone. White dashed lines highlight the transition of the original bones (left) to the newly formed bones (right). In this follow-up study, the tissues were collected at the end of the first (**A**), third (**B**), and sixth months (**C**), respectively. Blue arrows indicate active inflammation with the presence of exogenous compounds. Blue asterisk shows fibrous granulation tissue formation. Thin black arrows show sites surrounded by an increased number of osteoblasts. The thick arrow points to the multinuclear macrophage.

**Table 1 biomedicines-10-00662-t001:** Specific surface area, C-constant values, average pore diameters, cumulative pore volumes, and micropore contributions of the composite samples as a function of the sintering temperature.

Temperature	BET Surface Area (m^2^/g)	C-Constant	Average Pore Diameter (nm)	Pore Volume (cm^3^/g)	Micropore Volume (cm^3^/g)
500	400.0	82.46	26.34	3.397	0.026
600	415.0	56.33	33.98	3.254	0.021
700	382.8	64.81	33.97	3.251	0.000
800	384.9	59.40	31.95	3.074	0.021
900	277.0	51.18	46.73	3.237	0.005
1000	184.4	52.03	29.59	1.363	0.010

## Data Availability

Not applicable.

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
