# Peer review of "The Effect of Heat Treatment of β-Tricalcium Phosphate-Containing Silica-Based Bioactive Aerogels on the Cellular Metabolism and Proliferation of MG63 Cells"

_biomedicines, 2022, doi:10.3390/biomedicines10030662_

Round 1

Reviewer 1 Report

The manuscript contains extremely long sentences. It affects comprehension. The manuscript needs to undergo a thorough grammar, punctuation, and style check before it can be ready for publication. 

Line 20 - Rephrase and break down the information into smaller sections

Line 25 - Be specific about what kind of bone tissue

Line 27-30 - Needs to be rephrased and broken down

Line 38 - interest of needs to be changed to interest in 

Line 40 - Unclear

Line 46 - Grammar check

Line 69 - Use HAp to represent hydroxypaptite; HA may also mean hyaluronic acid

Line 75 - Grammar check

Line 89, 90 - What kind of temperature? It is unclear that the temperature here is the sintering temperature. 

Line 95 - Reference missing

Line 113 - Unclear

Line 139-142 - Unclear

Line 228/Figure 1 - Scale bar missing

It is unclear that the porosity was measured after the shrinkage. It would help to comment on the effect of sintering on reduction/increase in porosity.

Line 251 - Do you mean brittle? Instead of harder and rigid

Figure 3 - What is the significance of these SEM images? What is the reader looking at?

Figure 4 - The highest frequency pore happens to be around 20 nm. How does that compare to what is good for osteoblasts? Connect that observation back to what you see with cell growth.

Figure 5 - Reason out the trends seen here

Line 317-320 - Explain this observation.

Figure 8 - What is the reason for 600 and 700 showing low relative fluorescence? Does it have to do with the removal of chemical elements during the sintering process?

Line 371-374 - 700 shows lower cell viability but better collagen type 1 expression, how do you explain that? Could there be a correlation? Throw some light on it in the context of sintering temperature.

Line 398-420 - Watch capitalization. M in the word month should be in smaller cases. 

Line 478-481 - It is clear that the sintering affects cell behavior. But it looks like the lower sintering times work pretty much the same way as the other temperatures. The only difference between the samples appear to be in the histological study. It would be useful to comment on general rule of thumb for sintering temperatures for newer materials that may be developed in the future.

Reviewer 2 Report

The authors explore the dependence of amorphous and porous silica aerogel-based composites with β-tricalcium phosphate (β-TCP) as precursor  for hydroxyapatite formation in bone regeneration. The authors prepare, characterize and systematically test the β-tricalcium phosphate–mesoporous silica aerogel composites to reveal the dependence with the sintering temperature (600, 700, 800, 900 and 1000 °C). They employ the correct techniques for characterization:

  • The shrinking after temperature treatment was characterized by the cylinder´s diameter
  • The compressive strength was also tested
  • They present optical microscopy images of ground aerogel composites, pristine silica aerogel, and β-238 TCP after methylene blue selective staining of the silica matrix
  • Scanning electron microscopy images of the structure of aerogel matrices in the composites calcined at different temperatures
  • Surface properties, polarities, pore size distribution (Figure 4) and the internal void space (cumulative pore volume) were determined by nitrogen adsorption-desorption porosimetry
  • XRD result obtained on the as-prepared and annealed samples

Cell assays developed:

  • The chemical composition and concentrations of the leached elements (in ionic forms) were monitored for 5 days.
  • Molar ratios of the dissolved ionic species as a function of leaching time.
  • The effect on the viability or proliferation of the cells grown in the presence or absence (control) of the differently heat-treated aerogels was examined after 14 days of culture using Alamar Blue assay.
  • The alkaline phosphatase (ALP) activity of cells was measured after 14 days of incubation with the differently heat-treated aerogel composites
  • The gene expression of type 1 collagen in the cells grown in the presence or absence 370 (control) of the differently heat-treated aerogels was examined after 14 days of culture

The authors also add pictures from previous publication on the applicability of the monolithic mesoporous composite aerogel material sintered at the temperature of 1000 °C as a scaffold in rat calvaria (figure 11).

The article is well structured, the experiments are clearly exposed, and the results properly discussed.

Minor revisions:

  • Point 2.1. The previous published methodology should be referenced.
  • Figure 2. The c) is missing in the legend
  • Figure 11. In the legend NM is the abbreviation for new bone, while in the pictures appears NB
